# Violence against older women: a protocol for a systematic review of qualitative literature

Sarah R Meyer,[1] Molly E Lasater,[2] Claudia Garcia-Moreno[1]

[1]Department of Reproductive Health and Research, World Health Organization, Geneva, Switzerland
[2]Department of Mental Health, Johns Hopkins University Bloomberg School of Public Health, Baltimore, Maryland, USA

**Correspondence to**
Dr Sarah R Meyer;
smeyer@who.int

## ABSTRACT

**Introduction** There is sparse evidence globally concerning patterns of and types of violence against women aged 50 and older. Improved understanding of older women's experiences of violence, including types of violence, perpetrators and health impacts, is needed to address evident gaps in the literature, address requirements for monitoring and reporting on global sustainable development goal indicators, and inform policy and programming for preventing and responding to violence against older women. The aim of the systematic review is to identify, evaluate and synthesise qualitative studies from all countries, exploring violence against women aged 50 and above, identifying types and patterns of violence, perpetrators of violence and impacts of violence on various health outcomes for older women.

**Methods and analysis** A systematic search for qualitative studies of violence against older women will be conducted in the following databases: PubMed, PsycINFO, Embase, CINAHL, PILOTS, ERIC, Social Work Abstracts, International Bibliography of the Social Sciences, Social Services Abstracts, ProQuest Criminal Justice and Dissertations and Theses Global. Studies will be focused on violence against older women (aged 50 and above), using qualitative methodology, exploring women's experiences of any type of violence perpetrated by any type of perpetrator. Two authors will independently review titles and abstracts retrieved through the search strategy. Data extraction will be conducted independently by one author and quality assessment will be conducted by two authors, using an adapted version of the Critical Appraisal Skills Programme scale. Data will be analysed and synthesised using a thematic synthesis approach.

**Ethics and dissemination** Ethics approvals are not required as primary data are not being collected. Findings will be disseminated through a publication in a peer-reviewed journal and used to inform development of a module to measure violence against older women, for use in specialised violence against women surveys.

**PROSPERO registration number** CRD42019119467

## Strengths and limitations of this study

► This systematic review is designed with a comprehensive search strategy, to allow inclusion of all relevant qualitative studies of violence against older women globally.
► This review focuses on qualitative literature, in order to build understanding of lived experiences of older women subjected to different forms of violence.
► This systematic review uses established methods of systematic reviews of qualitative literature, including database searches, title and abstract screening, data extraction and data analysis.
► This review will use an adapted version of the Critical Appraisal Skills Programme scale to assess quality of studies.
► This review will use thematic analysis to synthesise findings from included studies and identify overarching themes and subthemes.

## INTRODUCTION

Violence against women (VAW) is a major public health problem, a gender inequality issue and a human rights violation. Violence has significant and long-lasting impacts on women's physical and mental health, including injuries, unintended pregnancy, adverse birth outcomes, abortions (often in unsafe conditions), HIV and sexually transmitted infections, depression, alcohol use disorders and other mental health problems.[1–5]

Much of this research has focused on women of reproductive age (15–49 years) as they suffer the brunt of partner violence and sexual violence. However, there is sparse evidence globally concerning patterns of and types of VAW aged 50 and older,[6] and this gap needs to be filled. A global research priority setting exercise on interpersonal violence indicated widespread consensus regarding the limited understanding of violence against older women. The exercise identified description of the nature, magnitude, distribution and consequences of elder abuse as priority research areas.[7] Specific risk and protective factors for violence victimisation among women of reproductive age may not be relevant or applicable in the case of older women. For older women, different relationship dynamics may influence forms of abuse,[8 9] recent exposure to violence may be interlinked with violence victimisation at different stages of the life course,[10 11] and

dynamics of ageing may influence decisions to disclose or report abuse.[12] Current lack of data on violence against older women may negatively impact service development and provision, including gaps in services, limited understanding of barriers to reporting and help seeking among older women who are subjected to violence. It may also lead to inappropriate policy or programmatic responses to violence.[10 13] International policy frameworks, including the Madrid International Plan of Action on Ageing specifically identifies elimination of violence, neglect and abuse of older women as a priority, recognising that older women's vulnerabilities are compounded by societal discrimination, poverty and lack of access to legal protections.[14]

There are currently two dominant theoretical frameworks to understanding violence against older women, each linked to different definitions of violence and assumptions regarding measurement, prevention of violence and other interventions: older adult mistreatment and older adult protection.[6 15 16] The older adult mistreatment framework is informed by social gerontology and understands violence as a form of elder abuse, focusing on age as the primary vulnerability to exposure to violence. The older adult protection framework specifically understands violence within the context of caregiving and institutional arrangements, where older adults' vulnerability to violence is a result of reliance on caregivers. There is also an approach that focuses specifically on one form of violence affecting older women, namely intimate partner violence (IPV). This approach adopts the definition of IPV used for women and girls below 50, and seeks to expand understanding of sexual, physical and psychological violence perpetrated by partners that older women may experience.

The lack of an overarching framework for understanding violence against older women has resulted in literature and evidence that is fragmented, with some research focusing only on specific types of violence against older women (eg, IPV),[17] some research (in particular, using the older adult mistreatment framework) lacking a focus on the gendered dimensions of violence,[16 18] and some approaches that focus specifically on older women in protective settings and relationships with caregivers rather than also including women in community settings. A 2013 United Nations report on neglect, abuse and violence against older women noted that these divergent theoretical frameworks have stymied data collection and evidence generation efforts; for example, some studies (in particular, population-based studies of IPV) have focused on women of reproductive age, while other studies (eg, studies of women in care settings) have excluded measurement of some types of violence, for example, perpetrated by intimate partners.[15] The elder abuse perspective has traditionally lacked understanding of the gendered nature of age-related vulnerabilities, for example, that women are less likely to have adequate pensions than men and that older women are more likely than older men to be financially dependent on family members.[6 19 20] Some researchers note that the term 'elder abuse' may homogenise 'older people rather than recognising individual differences, including gender'.[21] In the study of IPV, research has primarily focused on women of reproductive age, potentially leading to marginalisation of older women's experiences of IPV, which may differ in type and nature.[12] Researchers note that this conceptual split has resulted in 'lack of attention of ageing issues in research on VAW and, conversely, the lack of gender-based analysis in elder abuse research'.[22] Further, there is concern that these approaches primarily respond to the situation of older women in high-income, industrialised settings, and that forms of violence and vulnerability to violence more prevalent in low-income contexts are not adequately reflected in current literature, theoretical approaches and measurement methods.[15 22] Research is spread between disciplines and fields of inquiry, yet a more comprehensive understanding of violence against older women can be built by bridging these different approaches,[23] incorporating a perspective that is both age and gender responsive.[12] Improved understanding of older women's experiences of violence, including types of violence, perpetrators and health impacts, is needed to address evident gaps in the literature, address requirements for monitoring and reporting on global sustainable development goal (SDG) indicators, and inform policy and programming for preventing and responding to violence against older women.

This manuscript describes a protocol for a systematic review that complements a previous systematic review of quantitative studies of elder abuse (against men and women aged 60+). That review found the global prevalence of elder abuse in community settings of men and women is 15.7% in the past year, with psychological abuse and financial abuse the most prevalent forms of abuse reported. The studies were very heterogeneous and methodological variables were associated with prevalence, with larger sample size and random sampling associated with lower prevalence rates.[24] Sex was not found to be significantly associated with prevalence rates in the review, however, some studies did not provide sex-specific analysis. The prevalence of elder abuse in institutional settings may be as high as 64.2%, with data obtained from self-reports of perpetration by caregivers in institutions, with prevalence estimates highest for psychological abuse, and physical violence, neglect, financial abuse and sexual abuse less prevalent.[25] Findings from institutional settings indicate that being female is a significant risk factor for vulnerability to abuse. Analysis from the same review which focused on the findings specific to abuse of older women found a global prevalence of 14.1%, and similar to the analysis of men and women, psychological abuse was the most prevalent form of violence, followed by neglect. Financial abuse was less prevalent among women than in the analysis including both men and women.[22] Overall, there has been more work identifying and synthesising the quantitative literature on elder abuse and VAW,[22 26] which may miss out several forms of abuse

older women are exposed to, particularly in low-income and middle-income countries. This confirms a need to rigorously synthesise the qualitative literature.

There are some existing systematic reviews relevant to this one; those reviews primarily focus on specific forms of violence against older persons or women, and some focus on qualitative literature. Warmling *et al* conducted a systematic review of IPV against 'elderly men and women', focusing on prevalence studies (cross-sectional, population-based studies) in any country and exploring factors associated with the experience of violence by men and women.[18] The exact age range constituting 'elderly men and women' was not defined in the review. Findings included that psychological violence and economic abuse were the most prevalent forms of partner violence against older men and women, and alcohol use, depression, low income, functional impairment and previous exposure to violence were associated with this violence among older men and women. An empirical review of IPV in later life used a qualitative coding scheme to identify theoretical frameworks, conceptual themes and methodological approaches in the existing literature, examining 27 quantitative, 22 qualitative and 7 mixed-methods studies.[17] The review identified controlling behaviours and power dynamics in relationships that continue between intimate partners into later life, and may become further entrenched by caregiving dynamics. Similarly to the previous reviews, findings included that forms of IPV in later life shifted from a higher prevalence of physical and sexual abuse during reproductive age, to a higher prevalence of forms of psychological abuse, with results indicating 'a shift from physical to non-physical forms of violence dominated late-life scenarios.'[22] A review of qualitative research on IPV among older women identified a number of themes relevant to the dynamics of IPV against older women.[27] For example, women described patterns of abuse that were continuous and consistent with previous experiences of abuse in families of origin and previous relationships. However, patterns and types of IPV against older women were also described as shifting from predominantly physical violence experienced previously, to neglect, psychological violence and economic abuse becoming more prevalent. A systematic review and meta-synthesis of qualitative studies of IPV and older women focused on how previous exposure to IPV influenced health-seeking behaviours, specifically in the area of mental health care.[28] Bows conducted an empirical review of sexual violence against older people, including qualitative and quantitative studies focused specifically on sexual violence, with studies including any population of older adults (age range not defined), identifying widespread variation in prevalence rates across studies, with higher levels of sexual violence identified in studies using a domestic violence framework. The role of sociodemographic factors, such as ethnicity, marital status and living arrangements, was not consistently associated with violence victimisation, and studies identified a range of perpetrators, primarily intimate partners or

adult children.[23] A recent narrative review of IPV in later life included qualitative, quantitative and intervention studies focused on women aged 45 and above, finding that age and life transitions may result in older women experiencing IPV differently than younger women, facing unique risk factors and barriers to disclosure.[29] Among these existing systematic reviews of qualitative literature, none have focused specifically on older women, while being inclusive of any form of violence.

The present systematic review will build on previous systematic reviews and strengthen the evidence base by (1) focusing specifically on women, or if including studies with women and men, only including studies that provide sex-stratified analyses; (2) focusing on any form of VAW, rather than adopting a specific theoretical framework on what types of violence or perpetrators should be included from the outset (3) focusing on women aged 50 and above as many surveys often specifically focus on women up to 49 years of age and (4) focusing specifically on qualitative studies, to explore the nature and dynamics of violence against older women from the perspective of women and in low-income and middle-income countries.

## AIM

The aim of this systematic review is to identify, evaluate and synthesise qualitative studies from all countries, exploring violence against women aged 50 and above, identifying types and patterns of violence, perpetrators of violence, and impacts of violence on various health outcomes for older women.

## METHODS

The study was initiated in June 2018 and will be completed in June 2019.

### Search strategy

We identified the following domains as part of the research question: age (50 and above), women, violence and qualitative approach. For each of these domains, we identified the relevant keywords and search terms, which vary by database. Search strategy has been appropriately modified for each database, including syntax and specific terms, topics and/or headings.

We will not limit the search by year of publication, language of publication or type of publication at the stage of searching the databases. We have consulted with a librarian to provide input into the tailored search strategy that we have developed for each database, and the search strategy has been informed by other relevant systematic reviews and established approaches to identifying qualitative literature. The finalised search strategy for PubMed is included in online supplementary appendix 1.

We will also search reference sections of relevant existing systematic reviews to identify articles that fit inclusion and exclusion criteria. We have also identified experts in the field, including researchers, practitioners

and policy-makers, and have contacted them to provide any relevant literature. All experts will be contacted at least twice to provide the research team with additional resources to consider for inclusion.

## Data sources

We will conduct searches in the following electronic databases: PubMed, PsycINFO, Embase, CINAHL, PILOTS, ERIC, Social Work Abstracts, International Bibliography of the Social Sciences, Social Services Abstracts, ProQuest Criminal Justice and Dissertations and Theses Global.

## Data collection and analysis

### Eligibility criteria for the studies

The inclusion criteria for studies to be considered to be included in this review are:

### Inclusion criteria

1. Women aged 50 and older (studies including men or including women younger than 50 will be included if sex-specific and age-specific analyses respectively are included). This age range was selected because 15–49 years is generally considered reproductive age and most current data and evidence on IPV is in this age group.
2. Qualitative methodology (mixed-methods studies will be included if the qualitative data analysis is presented separately).
3. Studies focused on women's experiences of any type of violence perpetrated by any type of perpetrator.

There are no language or date limits. The abstracts of articles not in English will be reviewed by team members fluent in that language where feasible.

### Exclusion criteria

1. Whole sample is children or adolescents (under 50 years).
2. Sample of only men.
3. Only quantitative methodology.
4. Only includes perspective on VAW reported by care providers, health professionals, nursing home managers.

## Data management

We will use EndNote V.X7.8 as our bibliographic software management platform. We will remove duplicates using EndNote, prior to exporting titles and abstracts to an Excel spreadsheet for review. Data extraction and quality assessment results will be recorded in separate Excel spreadsheets. A flow diagram will be presented in any final publications, showing results of each stage of the review and adhering to the Preferred Reporting Items for Systematic Reviews and Meta-Analyses statement.

## Selection of studies

Two authors (SRM, MEL) will independently review titles and abstracts retrieved through the search strategy, to determine which should be included for full-text review. If an abstract or title is considered relevant by either of the authors, it will be included for full-text review. Two authors will independently review all articles selected for full-text review for eligibility, to reach consensus on inclusion in the review. Any discrepancies will be resolved with the input of the third author (CG-M). Reasons for excluding articles will be recorded.

## Data extraction

After full-text review, the following data will be extracted from all included articles using a standardised data extraction form: country studied, study design, research questions, sample size, characteristics of sample, data collection and analysis methods, main findings (as reported by the study's own authors), and any reported study limitations. Categories specific to the topic will also be included as data extraction progresses, including types and nature of violence, perpetrators of violence and reported impacts of violence.

Data extraction will be conducted independently by one author, and accuracy of the data extraction checked by a second author, with discrepancies resolved by discussion to reach consensus. Reviewers will develop and pretest a data extraction spreadsheet, to be used to compile a summary of characteristics and key findings of the included studies. The spreadsheet will also include categories relevant to data synthesis, described further below.

## Quality appraisal

### Risk of bias (quality) assessment

Included studies will be assessed for quality using an adapted version of the Critical Appraisal Skills Programme (CASP) scale. The adapted scale, relevant for review of qualitative studies, will include the following domains: research aims, methodology, research design, recruitment strategy, data collection, data analysis, reflexivity, ethical considerations, findings and research contributions.[30] The tool includes the following questions[31]:

1. Are the setting(s) and context described adequately?
2. Is the sampling strategy described, and is this appropriate?
3. Is the data collection strategy described and justified?
4. Is the data analysis described, and is this appropriate?
5. Are the claims made/findings supported by sufficient evidence?
6. Is there evidence of reflexivity?
7. Does the study demonstrate sensitivity to ethical concerns?
8. Any other concerns?

Two authors (SRM, MEL) will assess the quality of the studies and disagreement will be resolved either by discussion or inputs from the third reviewer (CG-M). Studies will be assessed as having an overall quality of 'high,' 'medium' or 'low' based on evaluation according to the CASP categories. Quality assessment will not be used to determine if any studies should be excluded, but rather to assess the strength of each study and as part of the confidence assessment (see below).

## Strategy for data synthesis

Findings will be presented and synthesised using a thematic synthesis approach. Following some descriptive analysis (eg, number of studies examining specific types of violence, which types of risks and outcomes are described in studies), two authors will use the findings extracted as part of the data extraction process to conduct initial open coding and develop broad themes to use to further sort the findings. The authors will use axial coding of the text units in order to develop first, second and third order themes.[32 33] The synthesis and all aspects of the systematic review process will be reported following the 21-item checklist provided in the Enhancing Transparency in Reporting the Synthesis of Qualitative Research statement.[34] If data allow, analyses comparing age groups (eg, 50–69 and 70+) and regions will be conducted.

## PATIENT AND PUBLIC INVOLVEMENT

Patients were not involved in the development of this systematic review. Public were not consulted specifically for the development of the research questions, however, previous research and consultations with experts have indicated that this is a fruitful and important area of research in the field of VAW research.

## DISCUSSION

This manuscript provides a description of a protocol for a systematic review of qualitative literature on violence against older women (aged 50 and above). Strengths of the proposed review include utilisation of multiple databases and search strategies to ensure inclusion of as much of the relevant literature as possible; a clear and structured process of data extraction and quality assessment; and transparent method of data synthesis, adhering to guidelines on systematic reviews of qualitative literature.[34] This systematic review of qualitative literature focuses on violence against older women, including any form of violence, and including studies conducted globally. It is an important complement to existing systematic reviews on the same and related topics, which have primarily included quantitative literature or focused only on IPV. One of the limitations of the review is that studies that do not include perspectives of women affected by violence themselves are excluded. There is a significant literature documenting the perspectives of caregivers and healthcare professionals on elder abuse and other forms of VAW, however, to ensure a focus on the lived experiences of women subjected to violence, we will exclude studies that only include perspectives of caregivers and healthcare professionals.

There are two SDG Indicators directly related to violence against women: 5.2.1, which measures intimate partner violence (Proportion of ever-partnered women and girls aged 15 years and older subjected to physical, sexual or psychological violence by a current or former intimate partner in the previous 12 months), and 5.2.2, which measures non-partner sexual violence) (proportion of women and girls aged 15 years and older subjected to sexual violence by persons other than an intimate partner in the previous 12 months). VAW prevalence studies, whether implemented as stand-alone or using a module as part of a Demographic and Health Survey, Reproductive Health or other type of survey, have primarily included women of reproductive age (15–49). Yet valid monitoring of and reporting on these SDG indicators requires data on older women to be collected and reported systematically. Some recent data collection efforts have sought to address this gap in data and evidence by including women aged 50 and above in studies using these standard approaches to measuring VAW (eg, the national prevalence studies in Bhutan and Laos[35 36]). However, this approach to measurement of violence against older women—using standard VAW survey measures—may not adequately capture the specific violence experiences of older women. Women aged 50 and above may experience different types and patterns of violence, with risk factors and impacts on physical and mental health, that differ compared with women aged below 50. This systematic review will inform efforts to increase and improve availability and quality of data on VAW aged 50 and above. This includes, for example, efforts led by WHO to develop or adapt a quantitative module to measure violence against older women. When we complete the systematic review, we will develop and refine a set of domains and themes that emerged from the data, to inform development of the module.

### Ethics and dissemination

We will present findings in a manuscript for publication in a peer-reviewed journal, through coordinated dissemination to researchers, practitioners, data users and generators with an interest in VAW, and to experts selected to participate in an expert meeting on violence against older women, to be convened by the Department of Reproductive Health and Research, WHO.

**Contributors** SRM and CG-M conceptualised and developed the systematic review. SRM drafted the manuscript. MEL and CG-M read and provided substantive feedback on all sections of the manuscript.

**Funding** The systematic review is funded by the Department for International Development, UNWomen-World Health Organization Joint Programme on Strengthening Methodologies and Measurement and building national capacities for Violence against Women data.

**Disclaimer** The funder had no role in developing the protocol.

**Competing interests** None declared.

**Patient consent for publication** Not required.

**Ethics approval** Ethics approvals are not required as primary data are not being collected.

**Provenance and peer review** Not commissioned; externally peer reviewed.

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
