## [Reviewer comments · BMJ Open]

ARTICLE DETAILS

TITLE (PROVISIONAL)	Violence against older women: A protocol for a systematic review of qualitative literature
AUTHORS	Meyer, Sarah; Lasater, Molly; Garcia-Moreno, Claudia

VERSION 1 - REVIEW

REVIEWER	Manuel Contreras-Urbina Global Women's Institute, George Washington University, Washington, D.C., USA
REVIEW RETURNED	28-Jan-2019

GENERAL COMMENTS	The manuscript "Violence against older women: A protocol for a systematic review of qualitative literature" presents the protocol the researchers will use to develop a systematic review on a very important topic related to gender-based violence (GBV). I would like to congratulate the authors for using a very rigorous methodology to develop a systematic review on such important topic. The protocol is developed following the strictest scientific standards for doing a systematic review and is explained in a very clear way. In addition, the systematic review will provide very important information on the situation of violence against a particular population that has been barely studied. The systematic review will fill a gap in knowledge. However, I am not totally sure that the methodology of how to conduct a systematic review, alone, is enough to be published in a high prestige peer-review journal. One option is that this information becomes part of the publication of the results of the review or it is published in a different source such a methodological report. I leave that decision to the editors.
--

REVIEWER	Hannah Bows Durham University, England, UK
REVIEW RETURNED	01-Mar-2019

GENERAL COMMENTS	This protocol is an important and timely proposal to conduct a systematic review examining qualitative studies addressing violence against older women. It will build on previous reviews and strengthen the evidence available in this field. My only suggestion/comment concerns the age range captured by the review. The authors propose a starting age of 50 for their inclusion
--

	criteria; although this is justified by the absence of women 50+ in existing research on violence against women, it is important to recognise that 50 and over incorporates multiple generations of women. Consequently, the analysis of the qualitative research included in this study will need to be sensitive to the differences between women at the young end (i.e. 50-65) and those at the mid and older ends. It is not always easy to do this as different studies adopted different starting points and some do not clearly define their use of the term 'older' or 'elderly', however it would provide important insights into the similarities and differences among the age groups which would have implications for policy makers and practitioners. Overall, this is a really promising protocol and it is fantastic to see further important work being done in this area.
--	--

VERSION 1 – AUTHOR RESPONSE

Many thanks to both reviewers for the supportive reviews and feedback. We agree with Reviewer 2 that analysis of different age groups is warranted if possible, and have added the following text under Data analysis to indicate this: "If data allows, analyses comparing age groups (for example, 50-69 and 70+) and regions will be conducted."